# Associations of Diet Quality and Sleep Quality with Obesity

**DOI:** 10.3390/nu13093181

**Published:** 2021-09-13

**Authors:** Soohee Hur, Bumjo Oh, Hyesook Kim, Oran Kwon

**Affiliations:** 1Department of Nutritional Science and Food Management, Ewha Womans University, 52, Ewhayeodae-gil, Seodaemun-gu, Seoul 03760, Korea; soohee1276@naver.com; 2System Health & Engineering Major in Graduate School, Ewha Womans University, 52, Ewhayeodae-gil, Seodaemun-gu, Seoul 03760, Korea; 3Department of Family Medicine, Seoul Metropolitan Government-Seoul National University Boramae Medical Center, 20, Boramae-ro 5-gil, Dongjak-gu, Seoul 07061, Korea; bumjo.oh@gmail.com

**Keywords:** Recommended Food Score (RFS), Pittsburgh Sleep Quality Index (PSQI), obesity, diet quality, sleep quality, sleep duration, effect modifier

## Abstract

Short sleep duration or poor sleep quality has been associated with an increased risk of obesity. Although the underlying mechanism remains unclear, one proposed pathway is poor diet quality. This cross-sectional study investigated whether diet quality modifies the association between sleep status and obesity in Korean adults. We used the baseline data and samples of 737 men and 428 women (*n* = 1165) aged 19–64, who participated in the prospective Ewha–Boramae cohort study. Sleep duration was dichotomized into ≥7 h (adequate) and <7 h (insufficient). Pittsburgh Sleep Quality Index (PSQI) values, reflecting sleep quality, were dichotomized into >5 (poor quality) and ≤5 (good quality). Diet quality was evaluated by the Recommended Food Score (RFS). Obesity was associated with higher rates of insufficient sleep and poor sleep quality in women, but not in men. After adjustment for covariates, women with poor sleep quality had a higher risk of obesity than women with good sleep quality (OR = 2.198; 95% CI = 1.027–4.704); this association occurred only in the group with RFS ≤ median score. Our findings support a significant association between sleep quality and obesity, and this association has been potentially modified by dietary quality in women.

## 1. Introduction

The prevalence of obesity has reached epidemic proportions across all genders, ages, and ethnicities. Short/inadequate sleep duration and compromised sleep quality are represented as contributing factors to the obesity epidemic [1,2,3]. Many epidemiologic studies, as well as meta-analyses and systematic reviews, have provided evidence that sleep status, such as sleep duration [4,5] and sleep quality [6,7], are associated with overweight and obesity.

The potential mechanisms linking sleep and obesity might include decreased physical activity and a low-quality diet [5,8]. Restricted sleep might affect food intake, appetite, satiety, and energy balance by modifying responses to hormones, such as leptin and ghrelin [9,10,11]. Moreover, poor sleep quality is often associated with unhealthy habits and lifestyle modifications, such as decreased physical activity and the consumption of high-calorie foods and beverages [12,13].

Findings from some studies of the national representative data of Korea indicated a causal relationship between short sleep duration and the increased consumption of dietary carbohydrates and/or carbohydrate-rich foods, which might lead to an increased risk of obesity [4,14]. In addition to the macronutrient intake status, the overall dietary quality index may be related to sleep status and obesity. To our knowledge, few studies have investigated the impact of dietary patterns and, especially, dietary indices on the relationship between sleep duration or quality and obesity. Therefore, the aim of this study was to determine whether diet quality modifies the association between sleep status and obesity.

## 2. Materials and Methods

### 2.1. Study Population

The study population included Korean men and women aged at least 19 years who had participated in the prospective Ewha−Boramae cohort study. All participants had undergone a full health assessment annually or biennially at Seoul National University Boramae Hospital (Seoul, Korea). Of the participants, approximately 86% had a free biannual health assessment under the Framework Act on Health Examination. The remaining participants had volunteered to take part in a private, noninsured health assessment.

The cross-sectional analyses in this study were conducted using the baseline data of 1465 participants (*n* = 889 men and *n* = 576 women) aged 19–80 years, which were collected between April 2015 and August 2016. After excluding the subjects who (1) were under 19 years and over 65 years (*n* = 116), (2) had missing data on the Recommended Food Score (RFS) and Pittsburgh Sleep Quality Index (PSQI) (*n* = 168), and (3) had missing data on the other covariates (*n* = 16); 1165 eligible participants were included (*n* = 737 men and *n* = 428 women). All subjects were then stratified by body mass index (BMI) into two groups: BMI < 25 kg/m^2^ (*n* = 416 men and *n* = 353 women) and BMI ≥ 25 kg/m^2^ (*n* = 321 men and *n* = 75 women).

The approval for this study was granted by the Institutional Review Board of Boramae Hospital (Approval number: 20140929/26-2014-118/102) and Ewha Womans University (Approval number: 86-8). All participants provided written informed consent following a comprehensive explanation of the procedures.

### 2.2. General Characteristics

Information on demographic characteristics and lifestyle, including smoking status, alcohol drinking status, and physical activity, was collected through self-administered questionnaires. Smoking status was categorized as never, past (defined as participants who quit over a year ago), or current smoker (defined as those who smoked at least 100 cigarettes in their life or smoked daily). Alcohol drinking status was categorized as non-current or current (defined as drinking more than once a month during the month before the survey). Physical activity was categorized as “yes” or “no”, depending on whether or not exercise was performed for ≥30 min/week. If the subject did at least 30 min of walking, moderate-intensity physical activity, or high-intensity physical activity during the past week, we classified the subject as “yes” in the physical activity category. Menopausal status was defined as “yes” or “no”, depending on whether menstruation had occurred for 12 months or more.

### 2.3. Anthropometric Parameters

Height and weight were recorded to the nearest 0.1 kg and 0.1 cm. Body weight was measured using a tetrapolar 8-point tactile electrode system (InBody 3.0, Biospace, Seoul, South Korea). Waist and hip circumference were measured according to the WHO guideline. BMI was calculated as weight (kg) divided by the square of the height (m^2^). Obesity was defined as BMI > 25 kg/m^2^, based on the WHO Asia-Pacific Area criterion for obesity [15].

### 2.4. Assessment of Sleep Duration and Quality

#### 2.4.1. Sleep Duration

Sleep duration was defined as the self-reported response to the question: “How many hours of sleep do you usually get in a day on average?” The responses were classified into two categories: short/inadequate sleep duration (<7 h/day) and adequate sleep duration (≥7 h/day), based on the recommended 7–8 h/day of sleep in previous studies [4,14,16] due to the increased risk of obesity in individuals with insufficient sleep, generally ≤6 h/day.

#### 2.4.2. Sleep Quality

The PSQI is a self-administered questionnaire that assesses subjective sleep quality during the previous month. This index comprises 19 items that evaluate seven components (range of subscale scores, 0–3): sleep quality, sleep latency, sleep duration, habitual sleep efficiency, sleep disturbance, use of sleeping medication, and daytime dysfunction. For example, for a sleep disorder, 0 = no sleep disorder, 1 = mild sleep disturbance, 2 = moderate sleep disturbance, and 3 = severe sleep disturbance. The sum of these seven component scores yields one global score for subjective sleep quality (range, 0–21); higher scores represent poorer subjective sleep quality, and a PSQI > 5 is associated with poor sleep quality [17].

### 2.5. Assessment of Diet Quality

The overall diet quality was assessed using the RFS developed by Kant et al. [18] and modified and validated for the Korean diet by Kim et al. [19]. Participants checked “yes” or “no” in response to at least weekly consumption of 46 recommended foods or food groups covering whole grains (1), nuts (1), tea (1), seaweeds (2), dairy products (3), legumes (4), fish (5), fruits (12), and vegetables (17). Each “yes” response was awarded 1 point. Participants received an additional 1 point if they reported eating three meals daily on a regular basis. Thus, the maximum possible score was 47 [19].

### 2.6. Blood Pressure and Blood Profiles

Systolic blood pressure and diastolic blood pressure were measured with a mercury manometer with the subjects in a sitting position. An average of two blood pressure readings at intervals of 5 min was used for the analysis.

Overnight fasting blood samples were collected in EDTA-containing tubes. Plasma was obtained by immediate centrifugation of the blood at 1500× *g* and 4 °C for 10 min (Allegra X-15R centrifuge, Beckman Coulter, Fullerton, CA, USA) and stored at −80 °C for subsequent carotenoid analysis. Total cholesterol, triglycerides, high-density lipoprotein, low-density lipoprotein, and fasting glucose levels were measured in whole blood using a Hitachi 7600 automatic analyzer (Hitachi, Tokyo, Japan). Plasma carotenoid levels (lutein, zeaxanthin, β-cryptoxanthin, α-carotene, and β-carotene) were determined by HPLC. The HPLC system (Shiseido Co. Ltd., Tokyo, Japan) was equipped with a photodiode array detector (Shiseido Co. Ltd.) and a YMC C30 reverse-phased column (5 μM, 4.6 × 250 mm; YMC Europe GMB, Dinslaken, Germany). The mobile phase consisted of a mixture of methanol:methyl *tert*-butyl ether (95:5 *v*/*v*) as solvent A, and methanol:methyl *tert*-butyl ether:1% ammonium acetate (8:90:2 *v*/*v*/*v*) as solvent B at a constant flow rate of 1 mL/min with the following gradient program: 95% A, 0.0–7.9 min; 90% A, 8.0–16.9 min; 55% A, 17.0–19.9 min; 42.5% A, 20.0–21.9 min; 90% A, 22.0–24.9 min, followed by an equilibration at initial conditions for 5 min. The total run time was 30 min. Chromatograms were monitored at 221 and 450 nm.

### 2.7. Statistics Analysis

All statistical analyses were performed using the SAS software (version 9.4; SAS Institute, Inc., Cary, NC, USA). The statistical significance was set at *p* < 0.05. Data are presented as mean ± standard deviation (SD) or number of subjects (%). Categorical variables were analyzed by Pearson’s chi-square test. Continuous variables were analyzed by the Student’s *t*-tests. To determine whether the diet quality (RFS median; 23 points for men and 21 points for women) modifies the association between sleep status and obesity, multiple logistic regression models were considered, with age, marital status, smoking, physical activity, and menopause status (women only) as covariates. The odds ratios (ORs) and 95% confidence intervals (CIs) for obesity were estimated with reference to sleep duration of ≥7 h/day and sleep quality (PSQI) score ≤ 5, respectively.

## 3. Results

### 3.1. General Characteristics of the Subjects According to BMI Group

The general characteristics of the two BMI groups (<25 kg/m^2^ vs. ≥25 kg/m^2^) are displayed in Table 1. The proportion of being classified as obese (BMI ≥ 25 kg/m^2^) was 43.6% for men and 17.5% for women. Compared to non-obese men, men with obesity showed higher proportions of married adults and current smokers, and compared with non-obese women, women with obesity showed higher proportions of older and menopause adults. The proportion of physical activity participation among men and women did not differ between the two BMI groups. When comparing general characteristics according to sex, the women were younger and had lower BMI, hip circumference, and waist circumference, and lower proportions of married people, current smokers, current drinkers, and physical activity participation compared to men.

Obese compared with non-obese women, but not men, had significantly higher rates of short sleep duration (<7 h/day) and poor sleep quality (PSQI > 5) (Table 2). Women had significantly lower sleep quality indicators (higher PSQI) and lower diet quality (RFS) compared with men. In both men and women, there was no difference in the dietary quality score (RFS) between the obese and non-obese groups.

The results of analyzing the correlation between RFS and blood carotenoid levels, known as antioxidant components, are shown in Appendix A. RFS was significantly positively correlated with levels of ß-carotene and total carotenoids in men, and lutein, ß-cryptoxanthin, α-carotene, ß-carotene, and total carotenoids in women. As a result of comparing blood profiles according to the RFS level (median), there was no significant difference in blood pressure, blood sugar, or other blood lipid-related indicators. When the confounding variables were adjusted, the blood level of β-carotene was significantly higher in the group with RFS above the median (RFS > 23) in men and women (RFS > 21), respectively, compared to below the median group of RFS (Appendix A).

### 3.2. Association between Sleep Status and Obesity According to RFS Median

Table 3 shows the association between sleep status and obesity by the RFS median resulting from multiple logistic regression analyses. Women with poorer sleep quality (PSQI > 5) had a greater risk of obesity (OR = 2.007; 95% CI = 1.008–4.281), even after adjusting for age (OR = 2.149; 95% CI = 1.027–4.494), and for age, marital status, smoking status, physical activity, and menopause status (OR = 2.198; 95% CI = 1.027–4.704); this association was found only in the group with RFS ≤ 21 (median value). In the men, when the association between poorer sleep quality and risk of obesity was analyzed according to the RFS median, there was no association in the unadjusted model, model 1, or model 2. There was no significant association between sleep duration and obesity with RFS in both men and women.

## 4. Discussion

To date, it is unclear whether the RFS, as a diet quality index, modifies the association between sleep duration or sleep quality and obesity. In this study, women with poor sleep quality (PSQI > 5) were found to have about twice the risk of obesity compared to women with good sleep quality (PSQI ≤ 5); this association was found only in the group with a diet quality lower than the median value (RFS ≤ 21) in the Korean adult population.

Although poor sleep health could increase the risk for obesity by increasing appetite and the consumption of a low-quality diet [4,14], to our knowledge, few studies have examined the relationship between sleep status and obesity in conjunction with dietary intake [20,21,22], and even less research has looked at diet quality indices [23]. A study of young Iranian women aged 18–28 years reported that diet quality indices (including dietary energy density, dietary diversity scores, healthy eating index, nutrient adequacy ratio, and mean adequacy ratio) among short sleepers (<6 h/day) were significantly lower than among longer sleepers [23]. In the current study, we found that the relevance between sleep quality and obesity only appeared in women with RFS lower than the median (i.e., women with relatively lower dietary quality). The RFS is derived by summing the consumption of antioxidant-rich foods and thus shows a positive relationship with plasma antioxidant status [24]. A positive correlation between RFS and the blood level of carotenoids, known as antioxidant components, was found in our study (Appendix A). We also found that the blood carotenoid level was significantly higher in the group with RFS above the median than that with RFS below the median (Appendix A). These results suggest that if RFS, an index reflecting dietary antioxidant status, is high, it may eliminate the inverse relationship between sleep quality and obesity risk.

On the basis of our results, short sleep duration or poor sleep quality was related to an increased risk of obesity, but only in women. Similar to our findings, some previous studies have also reported an inverse association between self-reported sleep duration and obesity exclusively in women [4,25]. According to Mezick et al., sleep duration and consistency were inversely related to body weight and body fat distribution; however, these correlations were stronger or exclusively evident in women [4]. Similarly, St-Onge et al. observed a stronger relationship between sleep duration and body composition in women than in men [26]. However, others have found that sleep duration [27] or sleep quality [28] was negatively associated with obesity in men but not in women. The source of this disparity is unclear, but it could be explained by gender-based differences in leptin and other metabolic hormones implicated in the sleep–wake cycle and food intake behavior in women [29,30]. In a review article on sleep and metabolic control, Trenell et al. highlighted that the sex/stress hormones, obesity, and aging all influence sleep duration/quality, either independently or in combination [31]. In a study of poor sleep quality, obesity, and anthropometric measurements of women, sleep quality had no significant relationship with BMI, waist circumference, triceps skinfold thicknesses, body fat ratio, or body fat mass [32].

We did not observe a direct relationship between sleep duration or quality with diet quality (data not shown). Substantial evidence [33,34,35,36] suggests that macronutrient and micronutrient intake are related to short sleep. However, few studies have examined the link between overall dietary quality indicators and sleep status rather than individual nutrients or food intake. Similar to our findings, Beebe et al. found no statistically significant variations in diet quality (Diet History Questionnaire II) between day and night shifts among nurses [37]. Meanwhile, Mondin et al. revealed that in Australian women, a diet quality score (DQS) was linked to adequate sleep duration (≥7 h) and a lower risk of short sleep duration (<7 h) [38].

Dietary variables, such as total energy and dietary composition, are recognized to play an important role in the etiology of obesity. Several epidemiological studies have suggested that protein and carbohydrate intake are linked to a decreased risk of obesity, but fat intake is linked to a higher risk [39,40]. However, this is still a contentious issue. Some carbohydrates, most notably fructose, are proposed to have a major role in increasing adiposity. The strongest link between a particular macronutrient and obesity, according to Riera-Crichton et al. [41], was carbohydrate; however, the type of carbohydrate was not specified. A high-fat/high-carbohydrate diet has been reported to cause obesity and if untreated leads to steatohepatitis [42]. In our study, the relationship between macronutrient intake and obesity could not be investigated because nutrient intake could not be calculated. When analyzing the relationship between diet quality (RFS) and obesity among diet-related indicators, there was no direct relationship between the two. Studies on the relationship between diet quality and obesity are inconsistent [43,44]. As such, the presence or absence of such a relationship may depend on which food quality indicator is used, and the race, lifestyle, employment status, family structure, socioeconomic status, general health status, and health risk behavior, particularly when sleep patterns are self-reported, rather than measured objectively.

The limitations of our study are as follows. First, the cross-sectional approach to this study means we cannot determine a causal relationship between diet quality and sleep status with obesity. Second, this study only used the RFS to assess the diet quality, and no instruments were used for evaluating the nutrients or intake amounts. Third, there may be unmeasured variables that result in unmeasured (or residual) confounding (e.g., socioeconomic status, depression, medication). Nevertheless, to the best of our knowledge, this is the first study to suggest that overall diet quality may modify the association between sleep quality and obesity in Korean adult women. This study has the potential to inform both policymakers and planners who are developing dietary guidelines about the role of diet quality or dietary patterns on obesity in adults.

## 5. Conclusions

In conclusion, a better diet quality might act as an effect modifier in the association between poor sleep quality and a high risk of obesity in women. This study suggests that high diet quality may play a beneficial role in obesity, which is associated with poor sleep quality in adult women. Future studies with larger sample sizes and a prospective or interventional design are needed to further improve our knowledge about the association of diet quality or dietary patterns and sleep quality with obesity.

## Figures and Tables

**Table 1 nutrients-13-03181-t001:** General characteristics of the subjects by body mass index (BMI).

Characteristics	Men				Women				
	Total (*n* = 737)	BMI < 25 kg/m^2^ (*n* = 416)	BMI ≥ 25 kg/m^2^ (*n* = 321)	*p*-Value *	Total (*n* = 428)	BMI < 25 kg/m^2^ (*n* = 353)	BMI ≥ 25 kg/m^2^ (*n* = 75)	*p*-Value *	*p*-Value **
Age (years)	46.8 ± 9.5	46.7 ± 9.8	46.8 ± 9.1	0.926	43.8 ± 10.1	43.0 ± 10.1	47.7 ± 9.4	<0.001	<0.001
Height (cm)	171.9 ± 6.4	171.7 ± 6.4	172.0 ± 6.4	0.606	159.9 ± 5.4	160.2 ± 5.6	158.8 ± 5.8	0.038	<0.001
Weight (kg)	73.0 ± 11.2	66.7 ± 7.5	81.2 ±9.7	<0.001	57.1 ± 9.2	54.3 ± 5.6	70.0 ± 11.8	<0.001	<0.001
BMI (kg/m²)	24.7 ± 3.2	22.6 ± 1.8	27.4 ±2.3	<0.001	22.3 ± 3.4	21.2 ± 1.9	27.7 ± 3.7	<0.001	<0.001
Hip circumference (cm)	95.8 ± 5.9	92.7 ± 4.4	99.8 ± 5.0	<0.001	92.3 ± 6.3	90.7 ± 4.5	99.7 ± 7.9	<0.001	<0.001
Waist circumference (cm) (cm)	87.0 ± 8.2	82.4 ± 5.9	93.0 ± 6.6	<0.001	79.2 ± 8.5	76.9 ± 6.3	89.9 ± 9.4	<0.001	<0.001
Married (%)	85.6	82.45	89.7	0.007	73.6	72.8	77.3	0.407	<0.001
Smoking (%)				0.011				0.477	<0.001
Non-smoker	38.3	42.5	32.4		94.6	94.6	94.7		
Former smoker	32	30.8	33.6		2.8	3.1	1.3		
Current smoker	29.7	26.4	34		2.6	2.3	4		
Current drinker (%)	64.7	62.7	67.3	0.229	48.1	47.3	52	0.541	<0.001
Physical activity ^†^ (%)	62	61.5	62.6	0.824	49.1	49	49.3	1	<0.001
Postmenopausal (%)					26.9	23.8	41.3	0.003	

Data are presented as mean ± SD or frequency (%). ^†^ Physical activity was defined as “yes” or “no”, depending on whether or not exercise was performed for ≥30 min/week. * *p*-value obtained by BMI 25 kg/m^2^ using *t*-test for continuous variables or chi-square tests for categorical variables. ** *p*-value obtained by sex using *t*-test for continuous variables or chi-square tests for categorical variables.

**Table 2 nutrients-13-03181-t002:** Sleep status and diet quality of the subjects by body mass index (BMI).

Characteristics	Men				Women				*p*-Value **
	Total (*n* =737)	BMI < 25 kg/m^2^ (*n* = 416)	BMI ≥ 25 kg/m^2^ (*n* = 321)	*p*-Value *	Total (*n* = 428)	BMI < 25 kg/m^2^ (*n* = 353)	BMI ≥ 25 kg/m^2^ (*n* = 75)	*p*-Value *	
Sleep duration									
Sleep duration (h)	6.7 ± 1.1	6.7 ± 1.1	6.6 ± 1.1	0.336	6.8 ± 1.2	6.8 ± 1.2	6.5 ± 1.2	0.104	0.150
Short sleep duration (<7 h/day) (%)	43.0	41.8	44.5	0.506	40.9	38.2	53.3	0.022	0.518
Sleep quality									
Pittsburgh Sleep Quality Index (PSQI)	5.8 ± 3.1	6.0 ± 3.3	5.9 ± 2.7	0.512	6.3 ± 3.7	6.2 ± 3.6	7.2 ± 3.9	0.003	0.006
Poor sleep quality (PSQI > 5) (%)	49.3	51.9	48.6	0.530	54.2	51.8	65.3	0.012	0.117
Diet quality									
Recommended Food Score (RFS)	23.4 ± 9.0	23.1 ± 8.8	23.9 ± 9.1	0.238	21.8 ± 8.5	21.7 ± 8.4	22.1 ± 9.1	0.673	0.002
RFS ≤ median ^†^ (%)	56.4	53.6	48.0	0.149	47.5	51.0	53.3	0.809	<0.001

Data are presented as mean ± SD or frequency (%). ^†^ RFS median values are 23 in men and 21 in women. * *p*-Value obtained by BMI 25 kg/m^2^ using *t*-test for continuous variables or chi-square tests for categorical variables. ** *p*-value obtained by sex using *t*-test for continuous variables or chi-square tests for categorical variables.

**Table 3 nutrients-13-03181-t003:** Association between sleep status and obesity by Recommended Food Score (RFS).

	Men		Women	
	RFS ≤ Median ^†^	RFS > Median ^†^	RFS ≤ Median ^†^	RFS > Median ^†^
	OR (95% CI) *	OR (95% CI) *	OR (95% CI) *	OR (95% CI) *
Sleep duration				
Unadjusted	1.020 (0.675–1.541)	0.830 (0.546–1.261)	1.983 (0.991–3.969)	1.691 (0.813–3.516)
Model 1	1.024 (0.677–1.548)	0.829 (0.546–1.260)	1.863 (0.919–3.776)	1.646 (0.783–3.461)
Model 2	1.019 (0.670–1.548)	1.288 (0.832–1.995)	1.921 (0.930–3.966)	1.627 (0.761–3.477)
Sleep quality				
Unadjusted	0.744 (0.493–1.124)	0.709 (0.466–1.077)	2.077 (1.008–4.281)	1.456 (0.689–3.075)
Model 1	0.746 (0.494–1.126)	0.703 (0.462–1.069)	2.149 (1.027–4.494)	1.390 (0.650–2.969)
Model 2	0.758 (0.500–1.150)	0.707 (0.458–1.090)	2.198 (1.027–4.704)	1.385 (0.636–3.016)

^†^ RFS median values are 23 in men and 21 in women. * Odds ratios (ORs) (95% confidence interval [CI]) were calculated with reference to long sleep duration (≥7 h/day) or good sleep quality (PSQI ≤ 5) using a multivariate logistic regression model. Model 1 was adjusted for age. Model 2 was adjusted for age, marital status, smoking, physical activity, and menopause status (women only).

## Data Availability

The data presented in this study will be made available by the corresponding author upon request.

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
