# Peer review of "Associations of Diet Quality and Sleep Quality with Obesity"

_nutrients, 2021, doi:10.3390/nu13093181_

Round 1
Reviewer 1 Report
Comments:
The study entitled as “Associations of Diet Quality and Sleep Quality with Obesity” (Manuscript ID# nutrients-1364806)” that high diet quality play a beneficial role in obesity, which is associated with poor sleep quality. Short sleep duration and poor diet may leads to obesity which is growing problem worldwide. 1,165 eligible participants were included in the current study with 737 men and 428 women. The effects are only seen in women population. Most studies reported no significant association between the Korean dietary pattern and obesity. The effort by authors in current study should be taken into consideration but there are still major issues need to be fixed before it can be published:
Comment 1: Line 45-56: High‐fat/high‐carbohydrate diet causes obesity and if untreated leads to steatohepatitis. (Kumar et al., Hepatology. 2020 Nov;72(5):1586-1604. doi: 10.1002/hep.31167.). There are chances that it may disturb sleep duration but alone high‐fat/high‐carbohydrate diet is sufficient to cause obesity. Authors should include opinion regarding this in the discussion section.
Comment 2: Line 153-154:” Obese compared with non-obese individuals of women, but not men, had significantly higher rates of short sleep duration (<7 h/day) and poor sleep quality (PSQI > 5)”( Table 2).” It is the result found by authors. Scientific justification for this should be included in discussion section.
Comment 3: Line 155-156: “In both men and women, there was no difference in the dietary quality score (RFS) between the obese and non-obese groups”. It means diet quality is not related with obesity in Korean adults?
Comment 4: Results section: Rewrite the results with more clarity. Results are described very briefly that is difficult to understand. Tables included in the result section should be explained by authors.
Comment 5: Table 1: Did physical activity not have any effect on obesity in studied Korean population?
Comment 6: Line 225-227: “A strong positive correlation between RFS and the blood level of carotenoids, known as antioxidant components, was found in our study (data not shown). It is interesting data and should be included in the manuscript.
Author Response
The study entitled as "Associations of Diet Quality and Sleep Quality with Obesity" (Manuscript ID# nutrients-1364806)" that high diet quality play a beneficial role in obesity, which is associated with poor sleep quality. Short sleep duration and poor diet may leads to obesity which is growing problem worldwide. 1,165 eligible participants were included in the current study with 737 men and 428 women. The effects are only seen in women population. Most studies reported no significant association between the Korean dietary pattern and obesity. The effort by authors in current study should be taken into consideration but there are still major issues need to be fixed before it can be published:
Reply) We sincerely appreciate the reviewer's insightful and constructive comments and suggestions. Please see our detailed responses below.
Comment 1: Line 45-56: High‐fat/high‐carbohydrate diet causes obesity and if untreated leads to steatohepatitis. (Kumar et al., Hepatology. 2020 Nov;72(5):1586-1604. doi: 10.1002/hep.31167.). There are chances that it may disturb sleep duration but alone high‐fat/high‐carbohydrate diet is sufficient to cause obesity. Authors should include opinion regarding this in the discussion section.
Reply) Thank you for the valuable comments. As suggested, we added a paragraph on the relationship between diet and obesity to the Discussion section. Additionally, we further described our study results in the context of the relationship between a high-fat/high-carbohydrate diet and obesity (L262-275).
Comment 2: Line 153-154:" Obese compared with non-obese individuals of women, but not men, had significantly higher rates of short sleep duration (<7 h/day) and poor sleep quality (PSQI > 5)" ( Table 2)." It is the result found by authors. Scientific justification for this should be included in discussion section.
Reply) Thank you for the valuable comments, which have helped us to improve the quality of the paper. As suggested, we have described in more depth the connection between sleep and obesity in women only in the Discussion section (L233-249).
Comment 3: Line 155-156: "In both men and women, there was no difference in the dietary quality score (RFS) between the obese and non-obese groups". It means diet quality is not related with obesity in Korean adults?
Reply) Yes, this study found that diet quality is not directly related to obesity. In this regard, we have reinforced and described this finding in more depth in the Discussion section (L259-275).
Comment 4: Results section: Rewrite the results with more clarity. Results are described very briefly that is difficult to understand. Tables included in the result section should be explained by authors.
Reply) As suggested, we rewrote the results more clearly and explained the data in the tables (L155-170).
Comment 5: Table 1: Did physical activity not have any effect on obesity in studied Korean population?
Reply) Table 1 is a simple comparison of general characteristics according to two BMI groups. The data provided is not conclusive evidence that physical activity does not affect obesity because it does not consider variables that may affect the relevance between obesity and physical activity (i.e., those that may act as covariates). Therefore, we described this aspect as simply a description of the table in the Results section (L155-156).
Comment 6: Line 225-227: "A positive correlation between RFS and the blood level of carotenoids, known as antioxidant components, was found in our study (data not shown). It is interesting data and should be included in the manuscript.
Reply) As suggested, the results of the correlation analysis between RFS and the blood level of carotenoids are presented as Supplementary Table S1 and are further described in the Results section (L162-165 and Table S1).
Reviewer 2 Report
The topic of the paper is interesting and the study is well designed and explained. There are only some concerns that I detail below:
Introduction: line 28-29: to erase have been represented and change it by are.
Materials and Methods: line 62: to change comma by semicolon before 1,165.
Table 1: to add cm after waist circumference.
Discussion: I think there is a mistake in line 186-187: It says: this association was found only in the group with a diet quality higher than the median value. (RFS>21). If I have understood rightly. It is should say diet quality lower than the median value (RFS <21). At least, this is what it described in the abstract and in results.
Line 229: Please make this sentence more clear: in the above the median RFS group compared to the below the median RFS group.
Conclusion: line 243: in my opinion you should avoid using it was confirmed and just say: In conclusion a better diet quality might act.....
Author Response
The topic of the paper is interesting and the study is well designed and explained. There are only some concerns that I detail below:
Reply) We sincerely appreciate the reviewer's insightful and constructive comments and suggestions. Please see our detailed responses below.
Introduction: line 28-29: to erase have been represented and change it by are.
Reply) As suggested, we changed it (L30). Thank you.
Materials and Methods: line 62: to change comma by semicolon before 1,165.
Reply) As suggested, we changed it (L64). Thank you.
Table 1: to add cm after waist circumference.
Reply) As suggested, we added it (Table 1). Thank you.
Discussion: I think there is a mistake in line 186-187: It says: this association was found only in the group with a diet quality higher than the median value. (RFS>21). If I have understood rightly. It is should say diet quality lower than the median value (RFS <21). At least, this is what it described in the abstract and in results.
Reply) Thank you for pointing out this critical oversight. As suggested, we changed it (L215). Thank you.
Line 229: Please make this sentence more clear: in the above the median RFS group compared to the below the median RFS group.
Reply) As suggested, we clarified the sentence (L228-230). Thank you.
Conclusion: line 243: in my opinion you should avoid using it was confirmed and just say: In conclusion a better diet quality might act.....
Reply) We agree with your comments. As suggested, we changed it (L287). Thank you.
Round 2
Reviewer 1 Report
Comments:
The revised version of study “Associations of Diet Quality and Sleep Quality with Obesity” (Manuscript ID# nutrients-1364806)” looks more promising and interesting. Authors tried to improve significantly the quality of manuscript by incorporating comments. One issue still need to fixed:
Comment 1: Results section: Rewrite the results with more clarity. Results are described very briefly that is difficult to understand. Tables included in the result section should be explained by authors. The authors only included the other raised comments in the result section but not elaborate the findings of tables.

Author Response
The revised version of study “Associations of Diet Quality and Sleep Quality with Obesity” (Manuscript ID# nutrients-1364806)” looks more promising and interesting. Authors tried to improve significantly the quality of manuscript by incorporating comments. One issue still need to fixed:
Comment 1: Results section: Rewrite the results with more clarity. Results are described very briefly that is difficult to understand. Tables included in the result section should be explained by authors. The authors only included the other raised comments in the result section but not elaborate the findings of tables.
Reply) Thank you for the comments. As suggested, we have rewritten the results so that they are clearer and better explained the data in the tables (L152-159; L180-183).